



# A Method for Preliminary Rotor Design - Part 1: Radially Independent Actuator Disk model

Kenneth Loenbaek[1,2], Christian Bak[2], Jens I. Madsen[1], and Michael McWilliam[2]

[1]Suzlon Blade Science Center, Brendstrupgaardsvej 13, 8210 Aarhus, Denmark
[2]Technical University of Denmark, Frederiksborgvej 399, 4000 Roskilde, Denmark

**Correspondence:** Kenneth Loenbaek (kenneth.loenbaek@suzlon.com)

**Abstract.** We present an analytical model for assessing the aerodynamic performance of a wind turbine rotor through a different parametrization of the classical Blade Element Momentum (BEM) model. The model is named the Radially Independent Actuator Disc model (RIAD) and it establishes an analytical relationship between the local-thrust loading and the local-power, known as the Local-Thrust-Coefficient and the Local-Power-Coefficient respectively. The model has a direct physical interpretation, showing the contribution for each of the 3 losses: wake-rotation-loss, tip-loss and viscous-loss. The gradients for RIAD is found through the use of the Complex-step-method and power optimization is used to show how easily the method can be used for rotor optimization. The main benefit of RIAD is the ease at which it can be applied for rotor optimization, and especially load constraint power optimization as it is described in Loenbaek et al. (2020). The relationship between the RIAD input and the rotor chord and twist is established and it is validated against a BEM solver.

**Keyword: BEM, Rotor Optimization**

## 1 Introduction

Wind turbine rotors are with their increasing size subject to continuous optimization with the overall objective of reducing the cost of energy. Such optimizations are very complex because both the aerodynamic and the structural performance need to be included in the optimization setup. Combining both aerodynamics and structural performance has shown very promising trends indicating that a further cost reduction is possible, see e.g. Perez-Moreno et al. (2016), Zahle et al. (2015), Bottasso et al. (2010). However, these optimization studies build on existing aerodynamic and aeroelastic tools, include numerous design variables and constraints, and can be very complex. Thus, it is challenging to make a very general optimization study to map the design space. That is the reason for investigating alternative methods and models to ease the exploration of optimal rotor designs.

The development of aerodynamic models for wind turbines is closely linked to that of propellers and helicopters. The first theoretical model for predicting the aerodynamic performance of a rotor was the so-called 1D momentum theory developed by Betz (Betz, 1926) and Joukowsky, which resulted in the famous maximum power extraction limit of $59.3\%$ known as the Betz-Joukowsky-limit or often just as the Betz-limit (Okulov and van Kuik, 2012). The model assumed constant loading along the rotor radius in the flow direction.





Later Glauert developed the Blade Element Momentum theory (BEM) (Durand and Glauert (1935), Sørensen (2016)), which
is an extension where momentum theory is used for radially independent stream-tubes. It also included correction models for
tip-loss and highly loaded rotors. The model has since then been extended with multiple correction models to account for yaw
misalignment, shear profiles, turbulent inflow, etc.

In this paper, we present an aerodynamic rotor performance model which we refer to as the *Radially Independent Actuator
Disc* (RIAD) model. It establishes a direct analytical relationship between the local-thrust loading and the local-power, which
is a useful simplification for rotor optimization. The model is equivalent to BEM but reduces the rotor design space to only two
independent variables at each radial station, i.e. the Local-Thrust-Coefficient ($C_{LT}$) and the glide-ratio ($C_l/C_d$) as well as the
global tip-speed-ratio ($\lambda$). The equivalent input for a BEM at each radial station is the design Lift-Coefficient ($C_l$), the design
Drag-Coefficient ($C_d$), and the Rotor-Solidity ($\sigma$) with the same global parameter. Most BEM formulations do not compute
the local-power directly, which is often an important optimization objective. At the same time, rotor optimization constraints
are often formulated in terms of loading. Both the objective and constraints are outputs from the BEM and their equations are
usually fairly convoluted. Using RIAD, the local-thrust loading ($C_{LT}$) is the independent design parameter and the local-power
is computed explicitly through a single equation. It makes it easy to recast the optimization problem, which generally requires
a robust optimization algorithm, into a straightforward root finding problem, which makes the optimization faster and more
robust.

The paper starts by presenting the derivation of the RIAD model, which then leads into computing the gradients for RIAD.
These gradients are used for power optimization, leading to a simple optimization method. In the end, the relationship between
RIAD inputs and blade chord and twist is then established, and RIAD is validated against a BEM solver. As this paper presents
an aerodynamic model to be used for optimization, this paper is Part 1 where the model is used for load constraint power
optimization in Part 2 (Loenbaek et al., 2020).

## 2    The RIAD model

In the following the *Radially Independent Actuator Disc* (RIAD) model is presented which start by establishing the relationship
between global and local parameters for a wind turbine rotor as well as introducing normalization. The relationship between
the local forces is then established, leading to an implicit equation for the local-power. A set of approximate closure equations
is then used to establish an explicit equation. The physical interpretation of the different factors and terms is then presented
and at the end, some details regarding the tip-loss-factor and the exclusion of drag from the induced velocity is discussed.





## 2.1 Relationship between global and local coefficents

Starting with the *Fundamental theorem of calculus* the following equation can be made for the global values for thrust ($T$) and power ($P$):

$$T = \int_0^R \frac{\partial T}{\partial r} dr \tag{1}$$

$$P = \int_0^R \frac{\partial P}{\partial r} dr \tag{2}$$

Where $\frac{\partial T}{\partial r}$ is the thrust-loading-density and $\frac{\partial P}{\partial r}$ is the power-density.

Introducing the classical non-dimensional relations for thrust ($C_T$) and power ($C_P$) as well as the local equivalent which is introduced as the *Local-Thrust-Coefficient* ($C_{LT}$) and *Local-Power-Coefficient* ($C_{LP}$):

$$T = \frac{1}{2}\rho_0 V^2 \pi R^2 C_T \tag{3}$$

$$P = \frac{1}{2}\rho_0 V^3 \pi R^2 C_P \tag{4}$$

$$\frac{\partial T}{\partial r} = \frac{1}{2}\rho 2\pi r V^2 C_{LT} \tag{5}$$

$$\frac{\partial P}{\partial r} = \frac{1}{2}\rho 2\pi r V^3 C_{LP} \tag{6}$$

Combining the equations the following equations can be found for $C_T$ and $C_P$:

$$C_T = 2\int_0^1 C_{LT}\tilde{r}d\tilde{r} \tag{7}$$

$$C_P = 2\int_0^1 C_{LP}\tilde{r}d\tilde{r} \tag{8}$$

Where $\tilde{r}$ is the Normalized-Radius ($\tilde{r} = r/R$). A sketch of the relationship between the local and global coefficients can be seen in figure 1.

## 2.2 Relationship between local coefficients

To establish a relationship between the local coefficients $C_{LT}$ and $C_{LP}$ the forces are assumed to be aerodynamic forces where the lift force is orthogonal to the local flow and where there is loss from viscous drag in the local flow direction. The force is assumed to originate from a rotating wind turbine blades with rotational speed $\omega$. The tangential force density is then given as $\frac{1}{\omega r}\frac{\partial P}{\partial r}$. Introducing the local lift density $\frac{\partial L}{\partial r}$ and drag density $\frac{\partial D}{\partial r}$ as well as the local flow angle $\phi$ the following equations can





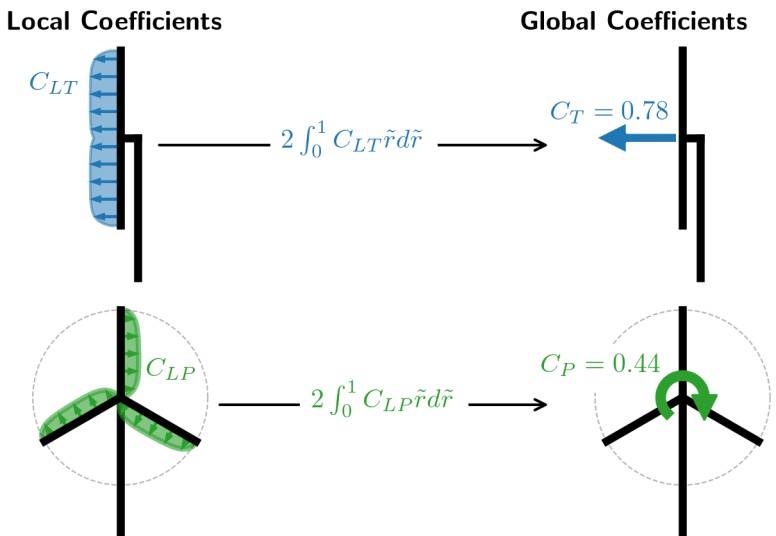

**Figure 1.** Sketch of the relationship between *Local Coefficients* ($C_{LT}, C_{LP}$) and *Global Coefficients* ($C_T, C_P$).

be made:

$$\frac{\partial T}{\partial r} = \frac{\partial L}{\partial r} \cos \phi \tag{9}$$

$$\frac{1}{\omega r} \frac{\partial P}{\partial r} = \frac{\partial L}{\partial r} \sin \phi - \frac{\partial D}{\partial r} \cos \phi \tag{10}$$

Where $\partial D/\partial r = 0$ in equation 9 to exclude drag from induction. Using the common normalization of $\partial L/\partial r$ and $\partial D/\partial r$

$$\frac{\partial L}{\partial r} = \frac{1}{2} \rho B c V_{rel}^2 C_l \tag{11}$$

$$\frac{\partial D}{\partial r} = \frac{1}{2} \rho B c V_{rel}^2 C_d \tag{12}$$

together with equation 5 and 6 the following equations can be found:

$$C_{LT} = \sigma \tilde{V}_{rel}^2 C_l \cos \phi \tag{13}$$

$$\frac{C_{LP}}{\lambda \tilde{r}} = \sigma \tilde{V}_{rel}^2 C_l \sin \phi - \sigma \tilde{V}_{rel}^2 C_d \cos \phi \tag{14}$$

Where $\sigma$ is the rotor solidity ($\sigma = Bc/2\pi r$) and $\tilde{V}_{rel}$ the normalized relative velocity. A sketch of the air-flow and forces can be seen in figure 2.

Combining equation 13 and 14 an equation for $C_{LP}$ can be found as:

$$C_{LP}\left(C_{LT}, \tilde{r}, \lambda, \frac{C_d}{C_l}, \phi\right) = \lambda \tilde{r} C_{LT} \tan \phi - \lambda \tilde{r} \frac{C_d}{C_l} C_{LT} \tag{15}$$

This is a general equation for the relationship between the $C_{LT}$ and $C_{LP}$, but it is implicit since $\tan \phi$ depends on $C_{LT}$.



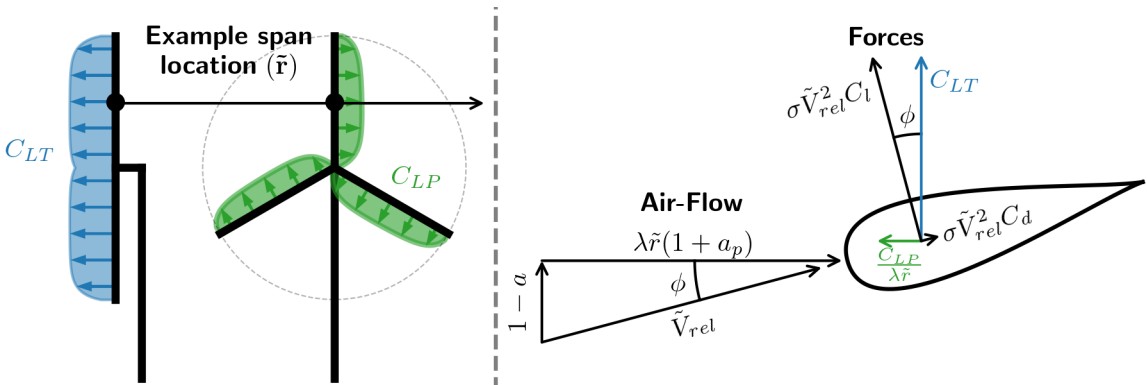

**Figure 2.** Sketch of the relationship between the air-flow and the forces at each span location.

### 2.3 Explicit equation for $C_{LP}$ (Closure relation between $C_{LT}$ and $\tan\phi$)

The local flow angle $\phi$ is given from the induced velocities (Sørensen, 2016, p. 101, eq 7.3) as:

$$\tan\phi = \frac{1-a}{\lambda\tilde{r}\,(1+a_p)} \tag{16}$$

where $a$ is the axial induction factor, $a_p$ is the tangential induction factor.

Equation 16 introduces two new variables $(a, a_p)$. In order to make an explicit equation for $\tan\phi$ as a function of $C_{LT}$ two equations relating $C_{LT}$ to $a$ and $a_p$ respectively are needed. These are the closure equations.

There does not exist a general set of model closure equations, but different approximate closures have been proposed. The most widely used set is referred to as the Glauert closure, which is an implicit assumption made for most BEM's. The closures are given as:

$$C_{LT} = 4a(1-a)F \qquad \text{(Ning, 2014, p. 4 eq. 2)} \tag{17}$$

$$a(1-a) = \lambda^2\tilde{r}^2 a_p(1+a_p) \qquad \text{(Sørensen, 2016, p. 50 eq. 4.36)} \tag{18}$$

Where $F$ is the tip-loss factor, which is further described in section 2.5. For now $F \in\, ]0,1]$ with $F \to 0$ as $\tilde{r} \to 1$. Combining equation 17, 18 with equation 16 an explicit equation for $\tan\phi$ in terms of $C_{LT}$ can be found. This leads to an explicit equation for $C_{LP}$:

$$C_{LP}\left(C_{LT}, \tilde{r}, \lambda, \frac{C_d}{C_l}\right) = \underbrace{\frac{1}{2}\left(1 + \sqrt{1 - \frac{C_{LT}}{F}}\right)C_{LT}}_{\text{1D power}} \cdot \underbrace{\frac{2\lambda\tilde{r}}{\lambda\tilde{r} + \sqrt{\lambda^2\tilde{r}^2 + \frac{C_{LT}}{F}}}}_{\text{wake rotation loss}} - \underbrace{\lambda\tilde{r}\frac{C_d}{C_l}C_{LT}}_{\text{viscous loss}} \tag{19}$$

Equation 19 is the main result of this section. It states an explicit relationship between the local power and the local loading.



### 2.4 Physical interpretation and input sensitivity

Equation 19 has a straightforward physical interpretation which is also highlighted through under bracing of factors and terms,

with an additional loss coming from the tip-loss factor ($F$). A sketch showing the impact of different losses for a specific input can be seen in figure 3.

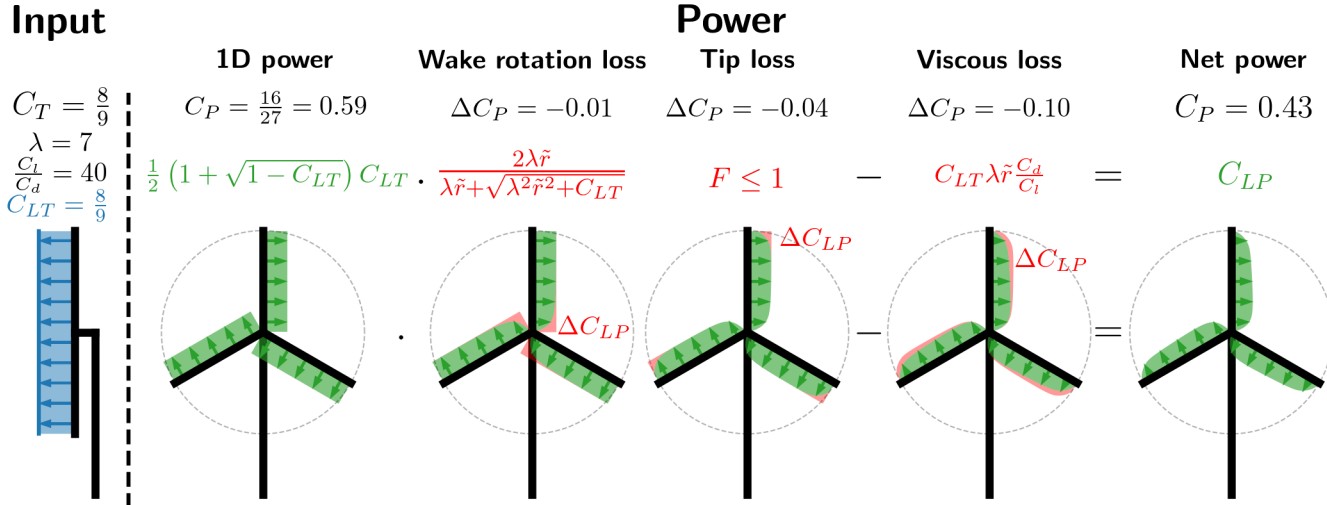

**Figure 3.** Sketch showing a graphical representation of the losses and the mathematical origin. The input is for span-wise constant local-thrust and glide-ratio.

The *1D power* is the classical 1D momentum theory result by Betz and Joukowsky (Okulov and van Kuik, 2012), but here applied for radially independent streamtubes $\left(\frac{1}{2}\left(1+\sqrt{1-C_{LT}}\right)C_{LT} = 4a(1-a)^2\right)$.

The *Wake rotation loss* is the power loss that originates from the conservation of angular momentum. When extracting power

from the rotational motion, the force that is rotating the blades leads to an opposing and equal magnitude force on the fluid, resulting in the fluid rotating in the wake of the turbine. Since there needs to be conservation of power, the potential power that can be extracted is lowered, leading to wake-rotation-loss. From figure 3 the wake-rotation-loss is seen to affect the root region of turbine. This is further investigated in figure 4, where the wake-rotation-loss-factor is plotted as a function of the local tip-speed-ratio ($\lambda\tilde{r}$) for different values of the local loading ($C_{LT}$). The effect of changing the local loading, is seen to have a

limited effect. From figure 4 b) the wake-rotation-loss is seen to be insignificant for $\lambda\tilde{r} > 5$ which is approximately from the mid-span and outwards for a modern utility scale turbine with $\lambda \approx 7-10$. From the example in figure 3 the wake-rotation-loss is seen to have the smallest impact on the global power with $\Delta C_P = -0.01$.

The *Tip loss* is the power loss associated with the rotor having a finite number of blades and not acting as an actuator disc with an infinite number of blades. This effect is captured in the tip-loss-factor ($F$) which is further described in section 2.5. The

120 tip-loss model that is used in this paper, captures the impact on the induced velocities in the rotor plane, with the additionally induced velocities coming from the vorticity released at the tip of the blade. The most important parameter for tip-loss, is the





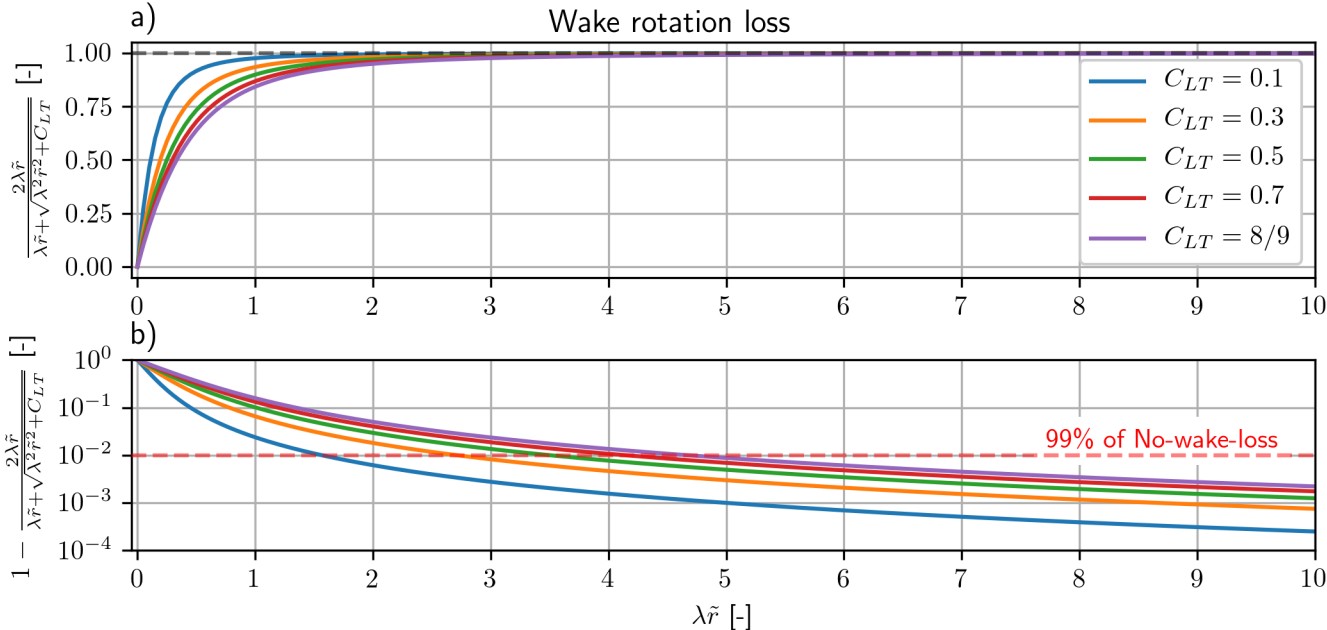

**Figure 4.** Significance of wake-rotation-loss. a) wake-rotation-factor vs. local-tip-speed-ratio ($\lambda\tilde{r}$). Black dashed line is the limit at $1/2$. b) difference between the wake-rotation-factor and the limit at $1/2$ vs. local-tip-speed-ratio ($\lambda\tilde{r}$). Notice that the y-axis is log-scale. Red dashed line is at 99% of the limit. Notice that the solid lines are for different values of $C_{LT}$. Higher $C_{LT}$ leads to higher wake-rotation-loss.

tip-speed-ratio ($\lambda$), with the tip-loss getting smaller with increasing tip-speed-ratio. From figure 3 the tip-loss is seen to affect the power at the tip as one might expect (hub or root loss was not included, but easily could have been).

The *Viscous loss* is simply the loss associated with the viscous drag from the airfoil profile. The viscous loss is found to be
linear in inverse-glide-ratio ($C_d/C_l$), loading ($C_{LT}$) and local-tip-speed-ratio ($\lambda\tilde{r}$). With a larger value for each of them leading to a larger loss. Opposed to both wake-rotation-loss and tip-loss a larger tip-speed-ratio is found to result in larger losses. As a consequence, there will exist an optimal tip-speed-ratio since either extreme ($\lambda \rightarrow 0, \lambda \rightarrow \infty$) will lead to $0$ or negative power. This is further discussed in section 3.3. From figure 3 the loss is seen to increase towards the tip since the local-tip-speed-ratio ($\lambda\tilde{r}$) is increasing. Viscous-loss is seen to be the most significant loss of the 3, although it should be noted that a glide-ratio of
$\frac{C_l}{C_d} = 40$ is a fairly low value for a realistic modern rotor design and is here chosen to make the loss easily visible for the figure.

## 2.5 Tip-loss factor

The tip-loss factor is commonly implemented for BEM's, and although some different tip-correction has been proposed, the tip-loss model by Glauert is a common one to used and it is also the one used here. It is given as:

$$F(\phi,\tilde{r},\lambda) = \frac{2}{\pi}\cos^{-1}\exp\left(-\frac{B(1/\tilde{r}-1)}{2\sin\phi}\right)$$
            (Sørensen, 2016, p. 132 eq. 8.29)        (20)



Where $B$ is the number of blades (which for simplicity is set to 3 thought out the paper). The Glauert tip-loss model leads to a recursive problem due to the mutual dependence between $C_{LT}$, $\sin \phi$ and $F$. It is not possible to find an explicit equation for $F$ in terms of $C_{LT}$, but it is possible to find a iterative scheme that can solve for $F$. The iterative scheme is given as:

$$\sin \phi_i \left( C_{LT}, \tilde{r}, \lambda, F_i \right) = \frac{1 + \sqrt{1 - \frac{C_{LT}}{F_i}}}{\sqrt{\left( 1 + \sqrt{1 - \frac{C_{LT}}{F_i}} \right)^2 + \left( \lambda \tilde{r} + \sqrt{\lambda^2 \tilde{r}^2 + \frac{C_{LT}}{F_i}} \right)^2}} \tag{21}$$

$$F_{i+1} \left( C_{LT}, \tilde{r}, \lambda, \sin \phi_i \right) = \frac{2}{\pi} \cos^{-1} \exp \left( -\frac{B(1/\tilde{r} - 1)}{2 \sin \phi_i} \right) \tag{22}$$

Where an acceptable tolerance for $F$ is reached with at most 30 iterations ($|F_{i+1} - F_i| < 10^{-9}$) where a good initial guess for $F$ would be 1.

The Glauert tip-loss model, breaks the explicit relationship between $C_{LT}$ and $C_{LP}$ in equation 19, since $F$ needs to be solved through iterations. An explicit relationship could be obtained by using the Prandtl tip-loss model instead, which is given as:

$$F_{Prandtl} \left( \tilde{r}, \lambda \right) = \frac{2}{\pi} \cos^{-1} \exp \left( -\frac{B}{2} \sqrt{1 + \lambda^2} (1 - \tilde{r}) \right) \qquad \text{(Sørensen, 2016, p. 131, eq. 8.26)} \tag{23}$$

Which does not depend on $C_{LT}$. Using Prandtl's tip-loss model, the effect on the tip-loss is found to be larger compared to Glauert's tip-loss model, but investigating it further here is out of scope for this paper.

Although the Prandtl tip-loss model is much simpler and easier to implement, the Glauert model is used throughout this paper as it is the one used for the BEM validation later in section 4.2.

**2.6   Blade loading without drag in induction**

Whether or not to include drag when computing the thrust loading (including drag in equation 13) is still a standing question for the derivation of BEM, and excluding drag in the derivation here should not be seen as an argument in favor of this approach, but merely as a consequence that it makes the mathematical derivation simpler as well as the resulting equation for local power (equation 19). If drag should be included for the computation of the induced velocities it should be noted that it is not just a

155 matter of including drag in equation 13, as the closure equation in equation 18 also needs an additional $\lambda \tilde{r} \frac{C_d}{C_l} (a + a_p)$ to be added on the right-hand-side, if it should be consistent with the BEM described in Ning (2014).

A consequence of excluding drag from equation 13 there arises a difference between the forces seen from the *blade*, and the forces seen by the *air* where equation 13 is the thrust force as seen by the air and where equation 14 is the tangential force seen by the blade. Often in optimization it is the thrust load at the blade that is of interest and equation 24 shows the relationship

between the two local-thrust loading's:

$$C_{LT,blade} \left( C_{LT}, \tilde{r}, \lambda, \frac{C_d}{C_l} \right) = C_{LT} \left( 1 + \frac{C_d}{C_l} \frac{1 + \sqrt{1 - \frac{C_{LT}}{F}}}{\lambda \tilde{r} + \sqrt{\lambda^2 \tilde{r}^2 + \frac{C_{LT}}{F}}} \right) \tag{24}$$

Where $C_{LT}$ is the local-thrust as seen by the air.



## 3  Gradients for RIAD and power optimization

In this section, a method for computing the gradients for RIAD is presented. The gradients are then used for power optimization.
First, it is applied for loading optimization for maximum power and it is then further extended for optimization wrt. tip-speed-ratio and loading. In the end, a discussion of how optimization with RIAD fits within the current state of the art is given.

### 3.1  Gradients with complex step

The local power equation (equation 19) is an analytical expression (with some complications for the tip-loss-factor) and in principle, it is straightforward to compute the gradient for any of the input variables. But it is tedious and error-prone and the tip-loss-factor makes it fairly complicated. The *Complex step method* (Martins et al., 2000) is therefore found to be an easier way to compute the gradients, without any loss of accuracy. It also has the benefit that little additional work needs to be done when equations 19 is implemented to compute the gradients. The conceptual idea behind the complex-step-method is fairly simple and for the sake of making the reader familiar with it, it is summarized here. For a proper description, the reader is referred to Martins et al. (2000).

The complex step method is based on the observation that the Taylor-series-expansion of an analytical function with a complex step (or perturbation) gives the following (taking equation 19 as an example with a step in $C_{LT}$):

$$C_{LP}\left(C_{LT}+ih,\tilde{r},\lambda,\frac{C_d}{C_l}\right) = C_{LP}\left(C_{LT},\tilde{r},\lambda,\frac{C_d}{C_l}\right) + ih\frac{\partial C_{LP}}{\partial C_{LT}}\left(C_{LT},\tilde{r},\lambda,\frac{C_d}{C_l}\right) + \mathcal{O}(h^2) \tag{25}$$

Where $i$ is the complex unit and $h$ the step size. Taking the imaginary component of equation 25 and dividing by the step size the approximate gradient can be found as:

$$\frac{\partial C_{LP}}{\partial C_{LT}}\left(C_{LT},\tilde{r},\lambda,\frac{C_d}{C_l}\right) = \frac{\mathcal{I}\left[C_{LP}\left(C_{LT}+ih,\tilde{r},\lambda,\frac{C_d}{C_l}\right)\right]}{h} + \mathcal{O}(h^2) \tag{26}$$

Equation 26 is seen to have some similarity to computing the gradient through finite difference. But the key difference is that finite difference requires the difference between two function evaluations, which leads to a rounding error. For finite difference there is an optimal step size ($h$) where the combination of the truncation error ($\mathcal{O}(h^2)$) and the rounding error is as small as possible. This is not the case for the complex-step-method, where the rounding error is eliminated by not computing a difference between two function evaluations and the step size can be arbitrarily small. Using a step size of $h = 10^{-9}$ the truncation error ($\mathcal{O}(h^2) \approx 10^{-18}$) is found to be smaller than machine precision ($10^{-16}$) and the gradient is therefore accurate to machine precision. This method applies to any analytical expression, but special care should be taken with functions that might lead to undesirable effects for complex numbers like the *absolute value function* or similar. This is not of concern for equation 19.



### 3.2 Loading optimization for max power

The problem of maximizing $C_P$ w.r.t. $C_{LT}$ is a classic problem, that will be used here to demonstrate how easily RIAD can solve this problem. It is thought to show the strength of using RIAD for optimization as opposed to using a regular BEM since the solutions is fairly easy to find.

The $C_P$ maximizing problem can be stated as:

$$\max_{\boldsymbol{C_{LT}}} \left[ C_P \left( \boldsymbol{C_{LT}}, \lambda, \frac{\boldsymbol{C_d}}{\boldsymbol{C_l}} \right) \right] \tag{27}$$

Where the bold-face signifies that it is a function/vector changing with span ($\tilde{r}$). Since the model assumes radial independence the maximization can be moved within the integration for $C_P$ (equation 8) and the maximization can be made for $C_{LP}$ at each span location ($\tilde{r}$) independently, which can be stated as:

$$\max_{C_{LT}} \left[ C_{LP} \left( C_{LT}, \tilde{r}, \lambda, \frac{C_d}{C_l} \right) \right] \tag{28}$$

Since equation 19 is a smooth function the optimization problem can be simplified as finding the root wrt. $C_{LT}$ for the following equation:

$$\frac{\partial C_{LP}}{\partial C_{LT}} \left( C_{LT}, \tilde{r}, \lambda, \frac{C_d}{C_l} \right) = 0 \qquad \text{for } C_{LT} \in \left[ 0, \frac{8}{9} \right] \tag{29}$$

$$\Downarrow$$

$$C_{LP,opt} \left( \tilde{r}, \lambda, \frac{C_d}{C_l} \right) = C_{LP} \left( C_{LT,opt}, \tilde{r}, \lambda, \frac{C_d}{C_l} \right) \tag{30}$$

Where $\frac{\partial C_{LP}}{\partial C_{LT}}$ is computed with equation 26, and the outcome from the optimization is the $C_{LT}$ that maximizes $C_{LP}$, which is referred to as $C_{LT,opt}$ and $C_{LP,opt}$ respectively. The root for equation 29 can be found with common root solving algorithms like bisection or Brent's method, eliminating the need for an optimizer to solve the problem. It should be noticed that to get the true gradient when including Glauert's tip-loss-factor (described in section 2.5), the complex step should be included when solving for the tip-loss-factor (equations 21, 22) otherwise the effect of the tip-loss on the gradient will not be included.

Applying the optimization with similar input as for figure 3 $\left( \lambda = 7, \frac{C_l}{C_d} = 40 \right)$, the resulting $C_{LT,opt}$ and $C_{LP,opt}$ is shown in figure 5. $C_{LT,opt}$ is seen to be mostly affected at the root and tip of the rotor, compared to the Betz-Joukowsky optimum. At the root $C_{LT,opt}$ is tending to a value of $3/4$ and at the tip going towards $0$. The mid-span is seen to have a slightly decreasing slope. For $C_{LP,opt}$ the 3 losses (wake-rotation-loss, tip-loss, viscous-loss) are highlighted as shaded region. $C_{LP,opt}$ is seen to have a local maximum at $\tilde{r} = 0.31$ from where viscous-loss and later tip-loss is seen to grow for increasing $\tilde{r}$. For decreasing $\tilde{r}$ the wake-rotation-loss is seen to increase.

### 3.3 Tip-speed-ratio optimization for maximum power

The *Loading optimization* in section 3.2 required two inputs $\left( \lambda, \frac{C_l}{C_d} \right)$ to maximize $C_P$, but in this section the optimization will be extended to also include the tip-speed-ratio as an optimization parameter leaving only the glide-ratio as an input. The





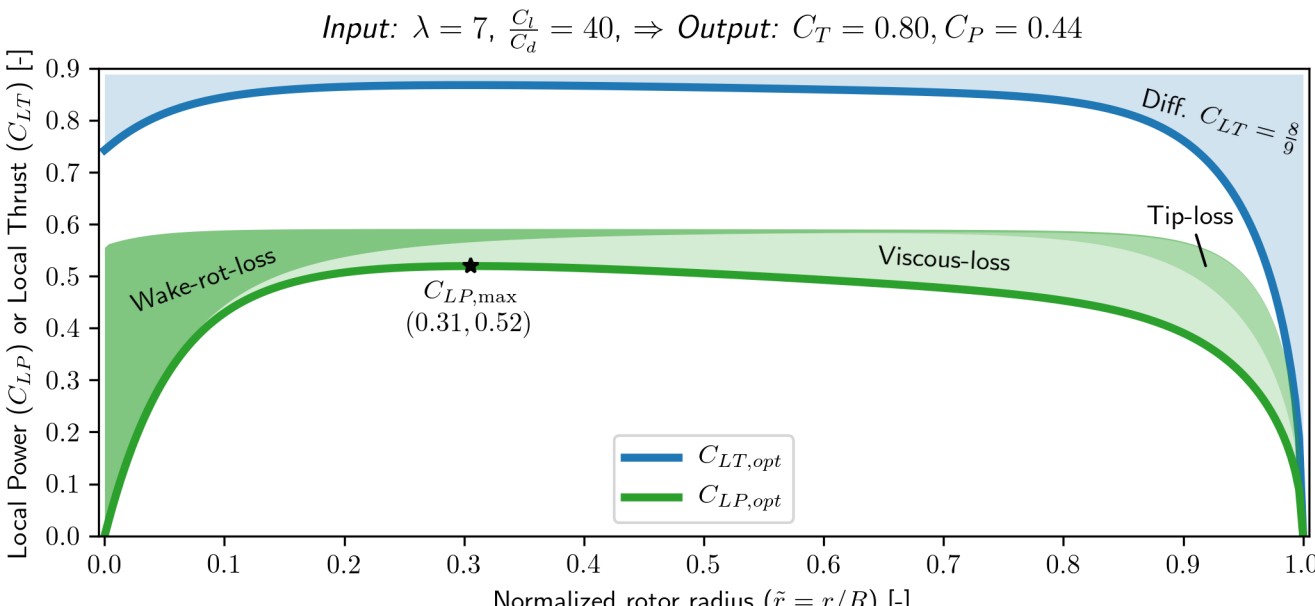

**Figure 5.** Optimal local-thrust ($C_{LT}$) and optimal local-power ($C_{LP}$) vs. normalized radius ($\tilde{r}$), with $\lambda = 7$ and span-wise constant $\frac{C_l}{C_d} = 40$ as input.

optimization problem can be stated as:

$$220 \quad \max_{\lambda, C_{LT}} \left[ C_P \left( C_{LT}, \lambda, \frac{C_d}{C_l} \right) \right] \tag{31}$$

Using the same assumption as for the loading optimization the optimization for $C_{LT}$ can be solved as it was described in section 3.2 assuming that $C_{LT,opt}$ is for a fixed $\lambda$. A nested optimization can therefore be stated as:

$$\max_{\lambda} \left[ C_P \left( C_{LT,opt}, \lambda, \frac{C_d}{C_l} \right) \right] \tag{32}$$

The solution to the above optimization problem can be found by solving:

$$225 \quad \frac{\partial C_P}{\partial \lambda} \left( C_{LT,opt}, \lambda, \frac{C_d}{C_l} \right) = 0 \qquad \lambda \in \left[ 0.2 \cdot \sqrt{\left( \frac{C_l}{C_d} \right)_{\min}}, \sqrt{\left( \frac{C_l}{C_d} \right)_{\max}} \right] \tag{33}$$

$$\Downarrow$$

$$C_{P,opt} \left( \frac{C_d}{C_l} \right) = C_P \left( C_{LT,opt}, \lambda_{opt}, \frac{C_d}{C_l} \right) \tag{34}$$

Where the bounding region is found from experience. The outcome from the optimization is the tip-speed-ratio that maximizes the power coefficient. They are referred to as $\lambda_{opt}$ and $C_{P,opt}$. To compute the gradient of $C_P$, equation 8 is used and the





complex step is applied as follows:

$$\frac{\partial C_P}{\partial \lambda}\left(\boldsymbol{C_{LT,opt}}, \lambda, \frac{\boldsymbol{C_d}}{\boldsymbol{C_l}}\right) = \frac{1}{h}\mathcal{I}\left(2\int_0^1 C_{LP}\left(C_{LT,opt}(r), \lambda + ih, \frac{C_d}{C_l}(r)\right)\tilde{r}d\tilde{r}\right) \tag{35}$$

Where for practical implementation the problem is discretized along the span and the integration can be performed using the trapezoidal rule. As it was the case for loading optimization the problem can be solved by the use of root-solving algorithms like bisection or Brent's method.

In figure 6 the result of solving the optimization problem for optimal tip-speed-ratio with varying span-wise constant glide-ratio is shown. As expected, the optimal tip-speed-ratio is seen to increase as the glide-ratio increase due to the balance between

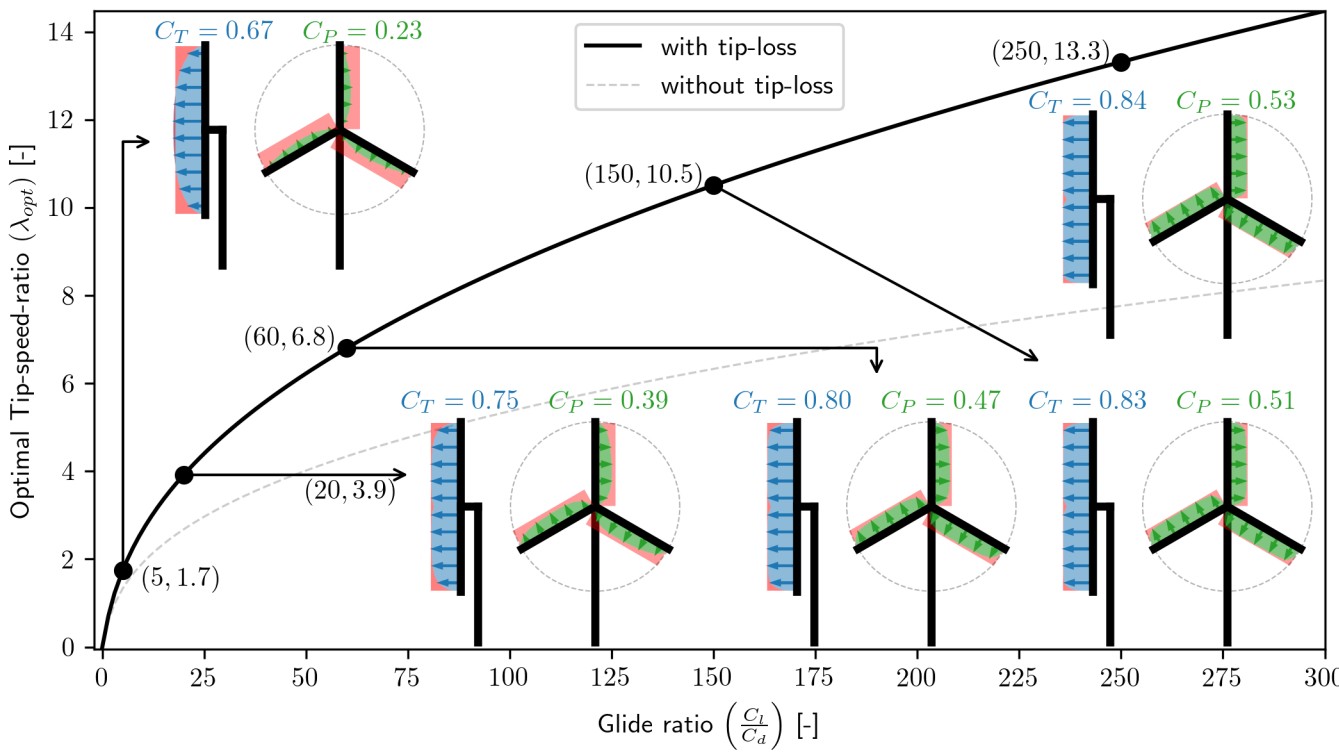

**Figure 6.** Optimal tip-speed-ratio ($\lambda_{opt}$) vs. span-wise constant glide-ratio ($\frac{C_l}{C_d}$). Both with and without tip-loss included in the optimization. For the case with tip-loss included, 5 points are highlighted, showing sketches of the local-thrust ($C_{LT}$) and the local-power ($C_{LP}$.

the viscous-losses (increases with increasing $\lambda$) and wake-rotation-losses as well as tip-losses (decreases with increasing $\lambda$). It is interesting to notice the significant impact of including tip-loss on $\lambda_{opt}$, with the inclusion of tip-loss leading to a larger $\lambda_{opt}$. 4 points along the $\lambda_{opt}$ curve in figure 6 is highlighted, showing sketches of the local-thrust and local-power with the
240 difference to the Betz-Joukowsky limit shown by shaded red region.

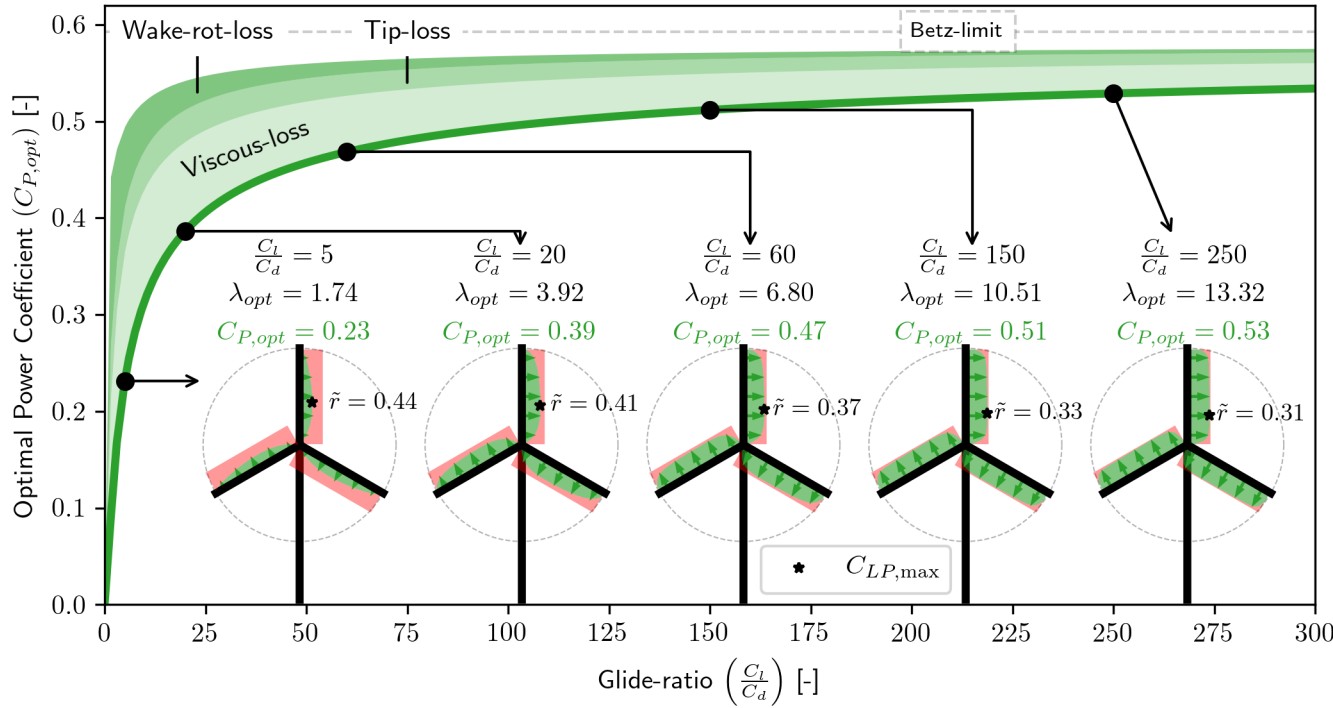

**Figure 7.** Optimal power coefficient ($C_{P,opt}$) vs. span-wise constant glide ratio ($\frac{C_l}{C_d}$). The same 5 points as in figure 6 is highlighted, but only the local-power ($C_{LP}$) is show for the sketches with the addition of the point of max local-power ($C_{LP,\text{max}}$).

The associated $C_{P,opt}$ is shown in figure 7. The 3 power loss contributions are shown as shaded regions and the viscous-loss is seen to be the most significant regardless of the glide-ratio. Tip-loss is seen to be the second most significant loss, at least for $\frac{C_l}{C_d} > 25$, which anyway would be a very low values for a modern utility scale wind turbine. The slope of $C_{P,opt}$ is seen to become flat for large values of the glide-ratio, and the improvement in $C_{P,opt}$ from $\frac{C_l}{C_d} = 100$ to $\frac{C_l}{C_d} = 150$ is $\Delta C_{P,opt} = 3.5\%$
and that is with a glide-ratio improvement of $\Delta \frac{C_l}{C_d} = 50\%$.

### 3.4 Compared to other work

The results of $C_P$ maximization presented in figure 5, 6 and 7 are not in itself novel results. Similar results have been shown by (Wilson et al., 1976, sec. 3.1-2), (Manwell et al., 2010, sec. 3.9), (Sørensen, 2016, cap. 5) and (Jamieson, 2018, sec. 1.9). The novelty is the ease at which these results can be obtained and the generality at which this method can be applied. In all of 250 the mentioned works, the optimization method relies on excluding mechanisms like rotational-effects, drag, or tip-loss to find a solution without the use of an optimizer. Common to them all is the exclusion of drag for the induced velocity as it was also done within this paper. However, the reason to exclude drag from the induced velocity within this paper is for the derivation of RIAD to be simpler. Including drag in the induced velocity will make equation 19 more complicated, but the optimization



methods presented in section 3.2 and 3.3 would still be applicable. A large part of why RIAD is easy to use and implement
for optimization is the use of the complex-step-method, which is arguably not an invention of RIAD and it could as well be
applied for a regular BEM, with the same result, although the optimization would be more convoluted. RIAD is established on
a better BEM parametrization dedicated to optimization when solving load-constrained power optimizations as it is shown in
Part 2 of this paper (Loenbaek et al., 2020).

## 4 From RIAD to rotor blade planform

Section 2.3 presented the connection between inputs, such as local-thrust, tip-speed-ratio and glide-ratio, for rotor power
performance and sections 3.2 and 3.3 presented the power optimization. But the presented methods only contains information
about the loading and power at the actuator disc, where this section establishes the connection between the RIAD inputs and
the blade planform, such as blade chord and twist. The blade planform is then used as an input for a BEM solver to validate
that RIAD and BEM are equivalent formulations.

### 4.1 Equations for chord and twist

An equation for chord can be found from equation 13 while applying the closure equations (equations 17, 18) for $\cos\phi$ resulting
in the following:

$$c(C_{LT}, \tilde{r}, \lambda, C_l, R, B) = \frac{8\pi\tilde{r}RC_{LT}}{BC_l} \frac{1}{\lambda\tilde{r} + \sqrt{\lambda^2\tilde{r}^2 + \frac{C_{LT}}{F}}} \frac{1}{\sqrt{\left(1 + \sqrt{1 - \frac{C_{LT}}{F}}\right)^2 + \left(\lambda\tilde{r} + \sqrt{\lambda^2\tilde{r}^2 + \frac{C_{LT}}{F}}\right)^2}} \quad (36)$$

To compute the chord it is seen that additional inputs are required such as lift-coefficient ($C_l$), the number of blades ($B$), and
270 rotor radius ($R$).

An equation for twist can be found in much the same way, by using equation 16 for $\tan\phi$ and applying the closure equations.
Combining it with: $\phi = \alpha + \theta_{twist} + \theta_{pitch}$, the following equation for the twist can be found:

$$\theta_{twist}(C_{LT}, \tilde{r}, \lambda) = \underbrace{\tan^{-1}\left(\frac{1 + \sqrt{1 - \frac{C_{LT}}{F}}}{\lambda\tilde{r} + \sqrt{\lambda^2\tilde{r}^2 + \frac{C_{LT}}{F}}}\right)}_{\phi} - \alpha - \theta_{pitch} \quad (37)$$

### 4.2 Validation with BEM

To show that RIAD is an equivalent formulation of the BEM equations, a planform design is created through equations 36 and
37 and evaluated with a BEM solver. The BEM solver used for the validation is *CCBlade* (Ning, 2014).

Running CCBlade requires an airfoil polar ($C_l, C_d$ vs. $\alpha$) and to keep it as simple as possible a single airfoil polar is used
all along the blade span. The airfoil is taken to be *FFA-W3-301* (Bjorck, 1990) with the aerodynamic data from Bak et al.
(2013). The design point for the polar was for simplicity taken as the angle-of-attack with maximum glide-ratio, resulting in





the following:

$$\frac{C_l}{C_d} = 92 \tag{38}$$

$$\alpha = 10.6° \tag{39}$$

$$C_l = 1.52 \tag{40}$$

Running the optimization for $\lambda_{opt}$ as described in section 3.3, gave $\lambda_{opt} = 8.4$ which is the tip-speed-ratio used for the design. To give the design some real dimensions, a rotor radius of $R = 50$m is used.

The resulting planform design can be seen in figure 8. Using the planform design as input for CCBlade as well as the other

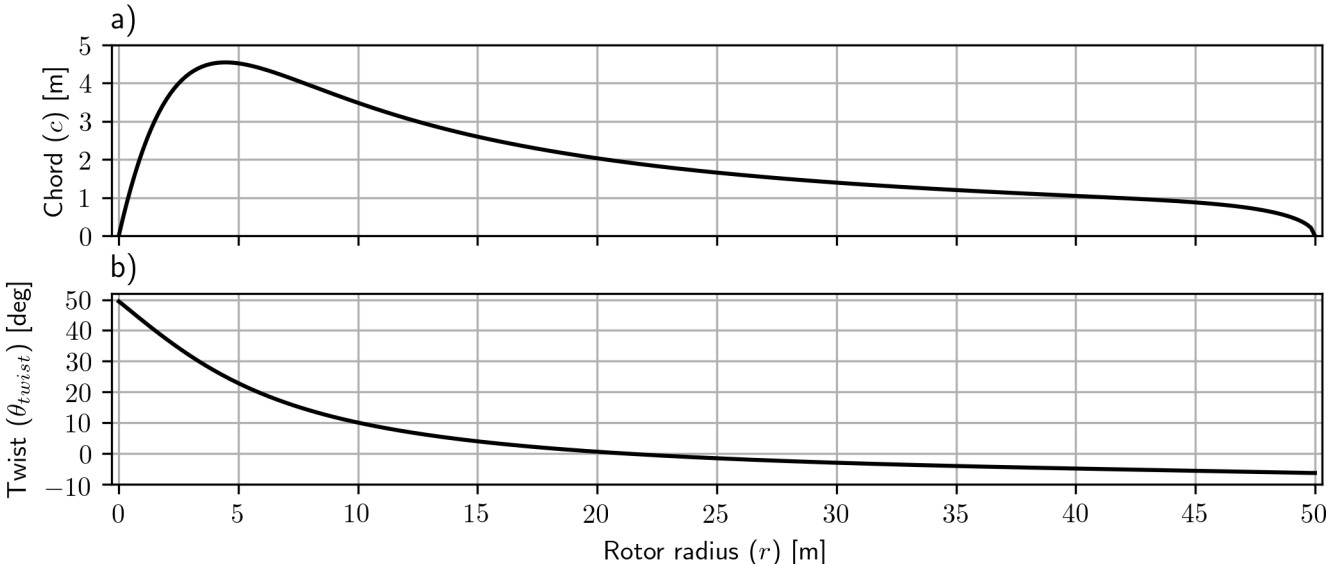

**Figure 8.** a) Blade chord b) Blade twist both as a function of radius. The number of blades is assumed to be $B = 3$.

inputs, the resulting local-thrust and local-power was found from CCBlade. A comparison between RIAD and CCBlade can be seen in figure 9. a) shows the values for $C_{LT}$ and $C_{LP}$ for both the solvers. b) shows the difference between RIAD and CCBlade for both $C_{LT}$ and $C_{LP}$ using a log-scale on the y-axis. Both as a function of the normalized radius. In the root region the two methods is seen to agree to machine precision, but with an error that is growing towards the tip and reaching a difference of $10^{-4}$ (which is still 3 significant digits). The growing error is found to disappear if tip-loss is excluded (agrees to machine precision) and the difference is also seen to disappear if the drag is included for the induction as well as including the tip-loss. The difference is therefore attributed to some small implementation difference regarding the tip-loss, but as the difference is anyway insignificant the difference is not investigated any further. It should be noted that for the comparison with CCBlade, it is $C_{LT,blade}$ that is used, where $C_{LT,blade}$ was discussed in section 2.6.

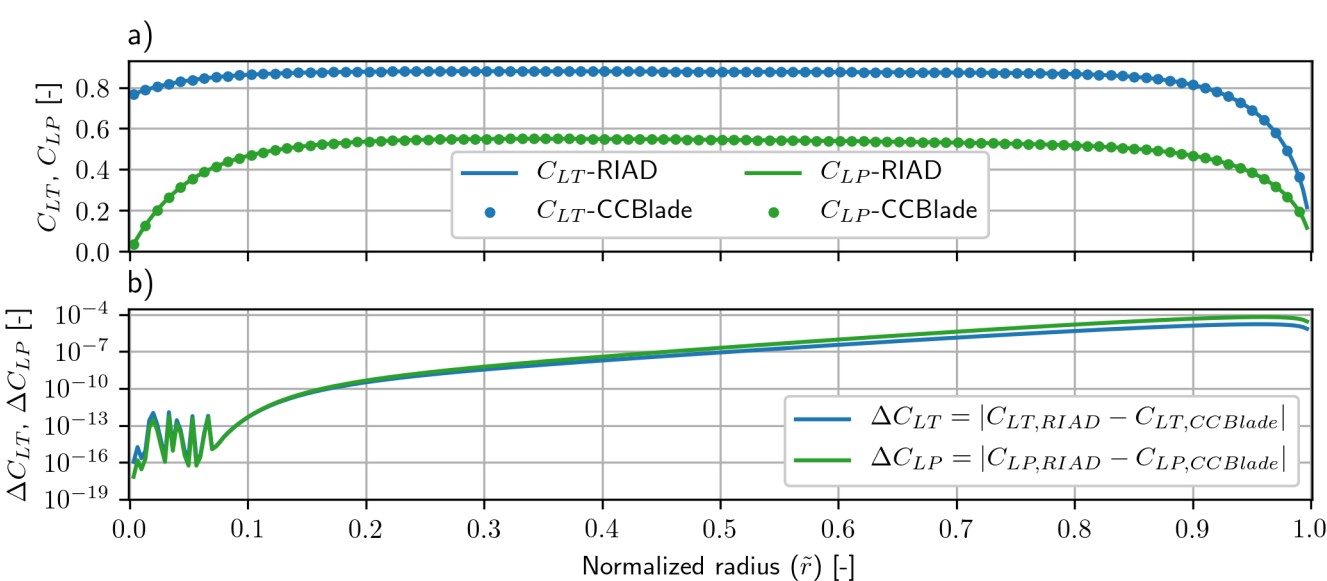

**Figure 9.** a) local-thrust ($C_{LT}$) and local-power ($C_{LP}$) for RIAD (line) and CCBlade (dots). b) the difference between the two methods for local-thrust ($\Delta C_{LT}$) and local-power ($\Delta C_{LP}$). The difference is seen to increase towards the tip, but the agreement is still within 3 significant digits and is therefore thought to be insignificant. The difference is likely a small implementation difference within tip-loss modeling.





# 5 Conclusion

A rotor performance model called *Radially-Independent-Actuator-Disc* model (RIAD) was presented. It is a different parametrization of the Blade-Element-Momentum (BEM) equations which is found to be better for wind turbine optimization. The model relates the local-rotor-power output (*Local-Power-Coefficient* - $C_{LP}$) to the local-rotor-loading input (*Local-Thrust-Coefficient*

- $C_{LT}$) at a given radial station ($\tilde{r}$). The model is a simple equation, shown in equation 19, from which different physical effects can easily be interpreted, such as wake-rotation-loss, tip-loss and viscous-loss.

A method to computing gradients for RIAD was presented, through the use of the *complex-step-method*, which allows to compute the gradient to machine precision with a minimum of additional work required.

The gradients were used for classical power-coefficient ($C_P$) maximization, which was first applied for loading optimization

($C_{LT}$ along the span) for a given tip-speed-ratio and glide-ratio. The optimization was then extended for combined optimization of tip-speed-ratio and loading, leading to a nested optimization for $C_P$ which only requires the glide-ratio along the span as an input. Using span-wise constant glide-ratio it was shown that viscous-loss is the most significant loss, regardless of the value of glide-ratio. The optimization results by themself have been done before, but the novel development is the ease at which the optimal result can be achieved and the generality at which the method can be applied. But the real strength of using RIAD for

optimization is for load-constraint rotor optimization as it is described in Part 2 (Loenbaek et al., 2020).

The relationship between local-thrust along the span and the blade chord and twist was presented and they were used to create the input for validation with a BEM solver (Ning, 2014). The difference between the two methods was found to agree to 3 significant digits with a likely small implementation difference for the tip-loss modeling. In this way, it was shown that RIAD and BEM are equivalent, with the difference being the parametrization.





# 6 Nomenclature

## 6.1 Rotor Local variables

Variables that are scalars at a given radius location ($r$). Bold-face variables indicates it is a function or vector changing with radius.

| Symbol | Description | Unit |
| --- | --- | --- |
| $\boldsymbol{x}$ | Bold face *local* variables symbolizes a function or vector changing with the rotor radius ($r$) | - |
| $r$ | Rotor radius variable $[0, R]$ | m |
| $\frac{\partial T}{\partial r}$ | Thrust loading density | $\mathrm{Nm}^{-1}$ |
| $\frac{\partial P}{\partial r}$ | Power loading density | $\mathrm{Wm}^{-1}$ |
| $\frac{1}{\omega r}\frac{\partial P}{\partial r}$ | Tangential loading density | $\mathrm{Nm}^{-1}$ |
| $\frac{\partial L}{\partial r}$ | Lift loading density | $\mathrm{Nm}^{-1}$ |
| $\frac{\partial D}{\partial r}$ | Drag loading density | $\mathrm{Nm}^{-1}$ |
| $\tilde{r}$ | Normalized rotor radius variable ($\tilde{r} = \frac{r}{R}$) | - |
| $C_{LT}$ | Local thrust coefficient (normalized $\partial T/\partial r$ - taken as the loading seen by the air) | - |
| $C_{LT,blade}$ | Local thrust coefficient as seen by the blade (including drag) | - |
| $C_{LT,opt}$ | Local thrust coefficient that maximizes $C_{LP}$ for a given $\lambda$ and $\frac{C_l}{C_d}$ | - |
| $C_{LP}$ | Local power coefficient (normalized $\partial P/\partial r$) | - |
| $C_{LP,opt}$ | Optimal local power coefficient for given $\lambda$ and $\frac{C_l}{C_d}$ | - |
| $a$ | Axial induction factor | - |
| $a_p$ | Tangential induction factor | - |
| $C_l$ | Lift coefficient | - |
| $C_d$ | Drag coefficient | - |
| $\frac{C_l}{C_d}$ | Airfoil glide ratio | - |
| $\frac{C_d}{C_l}$ | Inverse airfoil glide ratio | - |
| $c$ | Blade chord | m |
| $\sigma$ | Rotor solidity $\sigma = \frac{Bc}{2\pi r}$ | - |
| $F$ | Tip-loss-factor (described in section 2.5) | - |
| $\phi$ | Flow angle at the rotor plane (see figure 2) | deg |
| $V_{rel}$ | Relative wind speed $V_{rel} = \sqrt{(V(1-a))^2 + (\omega r(1-a_p))^2}$ | $\mathrm{ms}^{-1}$ |
| $\tilde{V}_{rel}$ | Relative wind speed $\tilde{V}_{rel} = \sqrt{(1-a)^2 + \lambda^2 \tilde{r}^2 (1-a_p)^2}$ | - |
| $\alpha$ | Airfoil angle-of-attack | deg |
| $\theta_{twist}$ | Blade twist | deg |



## 6.2 Rotor Global variables

Variables that are scalars for the whole rotor.

| Symbol | Description | Unit |
|---|---|---|
| $R$ | Rotor radius | m |
| $T$ | Rotor thrust | N |
| $P$ | Rotor power | W |
| $V$ | Free stream wind speed | $\mathrm{ms^{-1}}$ |
| $\theta_{pitch}$ | Blade pitch angle | deg |
| $\omega$ | Rotor rotational speed | $\mathrm{s^{-1}}$ |
| $C_T$ | Rotor thrust coefficient | - |
| $C_P$ | Rotor power coefficient | - |
| $C_{P,opt}$ | Rotor power coefficient with $\lambda_{opt}$ and $\boldsymbol{C_{LT,opt}}$ | - |
| $\lambda$ | Rotor tip-speed-ratio $\left(\lambda = \frac{\omega R}{V}\right)$ | - |
| $\lambda_{opt}$ | Rotor tip-speed-ratio that maximized $C_P$ for a given $\frac{C_l}{C_d}$ | - |
| $B$ | Number of blades | - |





*Author contributions.* KL came up with the concept and main idea, as well as made the analysis. All author have interpreted the results and made suggestions for improvements. KL prepared the paper with revisions of all co-authors.

*Competing interests.* The authors declare that they have no conflict of interest.

*Acknowledgements.* We would like to thank Innovation Fund Denmark for funding the industrial PhD project which this article is a part of.

   We would like to thank all the former employees at Suzlon Blade Sciences Center for being a great source of motivation with their interest in the results.

   We would like to thank Mads Holst Aagaard Madsen from DTU Risø for the inspiration to use the complex-step-method.



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
