# Peer review of "A Method for Preliminary Rotor Design - Part 1: Radially Independent Actuator Disk model"

_Wind Energy Science, 2020_

## Referee Comment (RC1) · Peter Jamieson (Referee) · 28 Oct 2020

Peter Jamieson (Referee)

peter.jamieson@strath.ac.uk

This paper describes a method, RIAD (Radially Independent Actuator Disc), for preliminary rotor design based on the usual assumption in blade element momentum (BEM) theories of radial independence of the blade elements and their associated annular rings of fluid. It is shown to be equivalent to BEM but by focusing on the primary variables power and thrust at each radial location via their associated local coefficients it is more insightful and simpler to implement top level rotor optimizations that may include load constraints.

I am very much a a fan of this kind of approach as all too often more complex sophisticated design tools are employed too early without the confidence that a design is going

in the best direction that could be gained from wider explorations at a higher level with simpler tools and analytic or semi-analytic methods.

The quality of the paper in general is very good and I have only some specific points to raise regarding a few details.

Last sentence of Section 1 Introduction reads a bit strangely with word "where". I understand that this paper is Part 1 (describing the method illustrated with power maximization) and Part 2 will deal with use for load constraint. Is that correct?

I really like Equation 4.19. Its very nice to see the power and loss terms clarified here.

On Figure 4 maybe wake rotation loss at top of figure a) should be deleted or amended. The title "Significance of wake-rotation loss is fine" but what you are showing as correctly stated in the expanded title below the figures in a) is the wake rotation factor with 1 corresponding to no loss and zero to maximum loss.

Regarding Section 3.4 - line 249; "The novelty is the ease ...". You referenced Jamieson 2018 just before that - the equation provided there for Cp max (for present large horizontal axis turbines with design tip speed ratio above 6 and max glide ratios around or above 100) will enable quite accurate estimation of Cp max without solving BEM equations which is about as easy as you can get! This assumes only that peak performance of each blade element is achieved at max glide ratio which is an excellent approximation but not quite exact.

You continue in Section 3.4 " In all of the mentioned work the optimization method.....". Just to be clear are you still referring specifically to the optimization method to determine Cp max or more generally? I have not checked the other references but formulae for optimum blade design and Cp max in Jamieson 2018 include drag, rotational loss via tangential induction factor and Prandtl tip loss factor. They are admittedly not fully optimized results because the tip loss in effect couples the elements. I think that sentence " In all of the mentioned work the optimization method....." may need to be

reworded or further explained.

"Common to them all is the exclusion of drag....". This is simply wrong for Jamieson 2018 if not for any of the others referenced. Drag is included in the induced flows and the equivalent of your equation 9 has additionally + dD/dr x sin phi.

Taking the paper as a whole I think the simplifications in dealing with local power and thrust coefficients and in using the gradient with complex step (with which I am not familiar) add up to a really nice way to do top level optimizations. The equivalence with BEM must hold since it is based on the same actuator disc theory and assumption of radially independent blade elements but it is good to demonstrate that analytically and computationally as a validation check.

---

## Referee Comment (RC2) · Anonymous Referee #2 · 2 Nov 2020

In the presented work, a novel method for the analysis of horizontal-axis wind turbines (HAWTs) is introduced. The proposed methodology, taking the name of RIAD (Radially Independent Actuator Disc), is based on a re-parametrisation of the Blade Element Momentum (BEM) method, currently adopted in turbine design and certification, in terms of distributions along the blade span of local thrust and power coefficient. As demonstrated by the authors throughout the paper, such strategy allows to re-formulate the performance assessment problem in a more physically sound way, clearly distinguishing the different aerodynamic loss contributions and their relationship with the turbine loads and power extraction. As a final result, the turbine preliminary design and optimization process is notably simplified.

The reviewer believes that the topic and the activity are very interesting, innovative

and worthy of investigation. The adopted methodology is rigorous and clearly detailed throughout the whole paper, which is very well presented.

Based on the aforementioned comments, the publication of the paper in the present form is strongly recommended.

---

## Referee Comment (RC3) · Anonymous Referee #3 · 23 Nov 2020

This paper is the first of a two-part series on Rotor Design. This paper deals with a radial independent actuator disk model (RAID). The actuator disk model is one of the most basic models used for describing either a wind turbine or propeller. The unique approach outlined in the paper is to look at a Local-Thrust-Coefficient and a Local-Power-Coefficient to quantify wake-rotation-loss, tip-loss and viscous loss. Using this approach will, according to the authors, lead to rotor optimization, the subject of Part 2. The paper introduction does a good job introducing the "why" of RAID. The derivation of CPT and CT is straightforward as is the description of CLP and CLT. The graphics, such as Fig 3, also make the paper easier to follow. Some developments are not so clear such as the CP,opt where the bounding region is found from experience (p11). This should be described in more detail. Describing the optimization of variables such

as tip-speed ratio, does make sense and is well presented. On page 13, I understand the reason to exclude drag but this is perhaps one of the more important factors in the design of the rotor and consideration should be given to including it in the model. I understand that the goal is to keep it simple but the model would be more useful if it wore included. For the validation with BEM, what is the Reynolds number used for the airfoil data? This is very important as far as airfoil performance. That would give some insight into the values used for cl/cd, alpha, and cl shown on p15. Figure 9 does show good agreement so what you assumed must be correct, if one believes the CCBlade values. Could this there be some experimental data with which to compare? In the end, this is a nice approach to rotor design however, I would have hoped that the paper could discuss in some additional detail why this approach is better that CCHelper or other design methods. What is the advantage?

---

## Editor Comment (EC1) · Alessandro Bianchini (Editor) · 26 Nov 2020

Dear authors, a number of interesting comments has been provided by the reviewers. Please carefully address all of them and reply at your earliest convenience. Best regards,

Alessandro

---

## Editor Comment (EC2) · Alessandro Bianchini (Editor) · 2 Feb 2021

Dear authors, reviewers' comments are in since a while. Please try to address them at your earliest convenience. Best regards
* * *

---

## Editor Comment (EC3) · Alessandro Bianchini (Editor) · 9 Mar 2021

Dear Authors, I have reviewed your responses to Reviewers' comments. Based on them, I encourage you to resubmit a revised version of your study that will be reconsidered for publication.

---

## Author Comment (AC1) · 9 Mar 2021

Dear Peter Jamieson,

Thank you for your very kind comments and for reading the manuscript carefully.

*Last sentence of Section 1 Introduction reads a bit strangely with word "where". I understand that this paper is Part 1 (describing the method illustrated with power maximization) and Part 2 will deal with use for load constraint. Is that correct?*
We understand that the sentence is confusing, and it has been updated as: This is Part 1 of a two-part paper. Part 1 describes an aerodynamic model for a wind turbine rotor and the use of the model for power optimization. Part 2 is described in Loenbaek

et al., 2020. where the model is applied for load constrained power optimization.

*On Figure 4 maybe wake rotation loss at top of figure a) should be deleted or amended. The title "Significance of wake-rotation loss is fine" but what you are showing as correctly stated in the expanded title below the figures in a) is the wake rotation factor with 1 corresponding to no loss and zero to maximum loss.*

We agree that the title of the figure is misleading, and the title has been changed from "Wake rotation loss" to "Wake rotation factor", which is thought to resolve the confusion.

*... the equation provided there for Cp max ... will enable quite accurate estimation of Cp max without solving BEM equations which is about as easy as you can get!*

The authors agree that the methods described in your work and the equation by Wilson are very simple and even though they are both approximate solutions they are fairly accurate. This is also the reason that we write "... the generality at which this method can be applied." as it is thought that the method described in our paper is more generally applicable, regardless of the choice of BEM equations.

*Just to be clear are you still referring specifically to the optimization method to determine Cp max or more generally? ...*

The authors do not clearly understand the difference between "CP max" and "more generally" in this context. We do see that the description is vague and not very specific. It is also recognized that the method presented in your work is not using an approximate equation as implied by the comment. We have therefore added a comment saying: "... or an assumption of constant axial induction." Which is thought to be an assumption of your work?! The others do apply some of these assumptions.

*"Common to them all is the exclusion of drag....". This is simply wrong for Jamieson2018 if not for any of the others referenced.*

Your work does indeed include drag for the induced velocity after carefully reading your work. The misunderstanding is that you do present an equation (eq. 1.79 p. 48) where drag is excluded from the induced velocity. Afterward, you then continue with

the more general equation where drag is included. We are sorry about the confusion and have updated the statement so that it is now only the 3 other references where drag is excluded from the induced velocity.

---

## Author Comment (AC2) · 9 Mar 2021

Dear Referee,

Thank you for the kind comments and a great summary of the work.

---

## Author Comment (AC3) · 9 Mar 2021

Dear Referee,

Thank you for your kind comments and for reading it carefully.

*Some developments are not so clear such as the CP,opt where the bounding region is found from experience (p11).*
We agree that it is not described in any detail how these limits are found. The range is mainly determined from a trial and error basis with some engineering judgment. We have changed from the previous vague sentence "Where the bounding region is found from experience." To the more descriptive: "Where the bounding region is found by observing that $\lambda_{opt}$ has an approximate proportional behavior of $\sqrt{\frac{C_l}{C_d}}$ and the limits

are simply determined to contain the optimal solution."

*On page 13, I understand the reason to exclude drag but this is perhaps one of the more important factors in the design of the rotor and consideration should be given to including it in the model.*
The considerations about including drag for the induced velocity is described in section 2.6 (Blade loading without drag in induction). It is thought to be too involved to present both cases and that it would lead to unnecessary confusion as the equation becomes much more convoluted. We have investigated the difference between including drag for the induced velocity or not in terms of $C_{P,opt}$ and the difference between including drag and not is within 99% for $C_l/C_d$>25. Showing this result is thought to be out of scope for this paper.

*For the validation with BEM, what is the Reynolds number used for the airfoil data?*
The Reynolds number does indeed play an important role in the aerodynamic performance in terms of Cl and Cd. CCBlade has the capability to use polars with different Reynolds numbers as the flow conditions change. Throughout this manuscript, it is assumed that the aerodynamic polars are fixed and hence they are the same for both RIAD and CCBlade. The value of the Reynolds number has been added for the input description.

*Could this there be some experimental data with which to compare?*
Comparing with experimental data or even just higher fidelity simulation methods (CFD) is of great importance for wind turbine design. With that said, it is thought to be out of scope within this paper to validate the BEM equations, hence why the focus has been to show that RIAD is equivalent to the classical BEM equations. A feature of RIAD is its ability to change the closure equations, and an expected part of our future work is to use higher fidelity method to test or make a corrected set of closures.

*... discuss in some additional detail why this approach is better that CCHelper or other design methods. What is the advantage?*

In section 3.4 (Compared to other work) we do mention how our work relates to work and methods by others. The main difference between RIAD and others is the parameterization of the BEM equations. RIAD is thought to be a better parameterization for optimization. We are not familiar with CCHelper and trying to search for it did not reveal any information, so it is hard to compare its difference and similarities.

---

## Author Response (AR1)

Dear Reviewers,

Thank you for reading and reviewing our manuscript carefully.

Based on your comments we have made the following main changes (see the diff document for the changes):

- (p. 2 line 43) The sentence describing that the paper is a two-part paper has been updated as it was confusing
- (p. 11 line 233) The explanation for setting the range for the CPopt was been updated with further explanation.
- (p. 13 line 258) The statement of the authors who exclude drag has been to reflect that Jamieson do indeed include drag.